# Review of Nonpoint Source Pollution Models: Current Status and Future Direction

Mingjing Wang [1], Lei Chen [1,*], Lei Wu [2], Liang Zhang [3], Hui Xie [4] and Zhenyao Shen [1]

1   State Key Laboratory of Water Environment Simulation, School of Environment, Beijing Normal University, Beijing 100875, China
2   Key Laboratory of Agricultural Soil and Water Engineering in Arid and Semiarid Areas, Ministry of Education, Northwest A&F University, Xianyang 712100, China
3   Innovation Academy for Precision Measurement Science and Technology, Wuhan 430077, China
4   Nanjing Institute of Geography and Limnology, Chinese Academy of Sciences, Nanjing 210008, China
*   Correspondence: chenlei1982bnu@bnu.edu.cn

**Abstract:** Modelling tools are commonly used for predicting non-point source (NPS) pollutants and it is timely to review progress that has been made in terms of the development of NPS models. This paper: (1) proposes a systematic description of model framework and generalizes some commonly used models; (2) identifies the common challenges in model structure and applications; (3) summarizes the future directions of NPS models. Challenges in model construction and application are based on the following: (1) limitations in understanding specific NPS pollution processes; (2) model expansion to different scales; (3) data scarcity and its impacts on model performance; (4) prediction uncertainty due to model input, parameter and model structure; (5) insufficient accuracy for decision-making. Finally, this paper proposes future directions for model development, including: (1) a source–flow–sink framework for model development; (2) standardization for model input and parameter; (3) reliable decision support for environmental management. The findings of this review provide helps in the accurate prediction and management of NPS pollution around the world.

**Keywords:** nonpoint source pollution; model construction; model application; uncertainty; decision-making; best management practice




## 1. Introduction

The rapid development of urbanization and agriculture has accelerated nonpoint source (NPS) pollution that has led to the deterioration of surface water quality [1]. Simulating the characteristics and the pathways of NPS pollutants would help to identify their hot spots and environmental impacts to formulate region-specific controls and improve the management efficiency [2]. However, it is a large challenge to track NPS pollutants from production to final fate due to their various forms and multiple paths [3]. Actual measurement data are important for model calibration and verification. Additionally, most modeling even requires ongoing collection of data since calibrations/relationships change. However, the stochasticity, indirection, and uncertainty of NPS pollutants emission along the land–water continuum, as well as the temporal and spatial variability, make it difficult for their measurement and simulation [4,5].

As an effective instrument, models have been widely used in the prediction and management of NPS pollution in many watersheds around the world [6,7]. However, there are challenges in NPS models, such as the massive, measured data required for model, uncertainties in model application, insufficient accuracy for decision-making and so on. Given the impact of NPS pollution and the fact that these impacts are likely to increase in the future, and the difficulty of assessing the relative benefits of different NPS-reduction options, it is timely to review progress that has been made in terms of the development of NPS models and to identify future research directions.

Reviews of NPS models in the last two decades have focused on (1) model identification and provide suggestions for model selection [8]; (2) reviews for models applicable to particular geographic scopes, such as catchment, field, and urban areas [7,9]; (3) the development and application of a single model [10]; (4) for particular pollutants, such as phosphorus (P), nitrogen (N) or sediment [11]; and (5) other aspects for modeling, such as the Best Management Practice (BMP) scenario, uncertainty analysis, and sensitivity analysis [12,13]. This review focuses on the following: (1) proposed a systematic description of model framework and generalized some commonly used models; (2) identified the common challenges in model structure and applications; (3) summarized the future directions.

## 2. Overview of NPS Model

### 2.1. Basic NPS Pollution Processes and Model Framework

NPS pollution involves multi-spherical migration and transportation process that are influenced by various natural conditions and human activities [14]. Due to the inconsistency of underlying surfaces, the terrestrial and aquatic transport processes are significantly distinct, but both are complex [7,15]. Pollutants on the terrestrial surface usually move with runoff and converge into rivers or lakes. Meanwhile, with the evaporation, infiltration, and interception of vegetation, some pollutants enter the groundwater or are deposited. Among these processes, pollutants may be adsorbed or desorbed by soil/sediment and banks, deposited or resuspended, while reactions occur between pollutants in the water column and they are influenced by plant or aquatic organisms. In wetland or lakes, the retention time is long due to the slow velocity of water flow, which is conducive to the accumulation of NPS pollutants; in addition, lake stratification causes the migration and transportation of pollutants different from those of rivers (Figure 1).

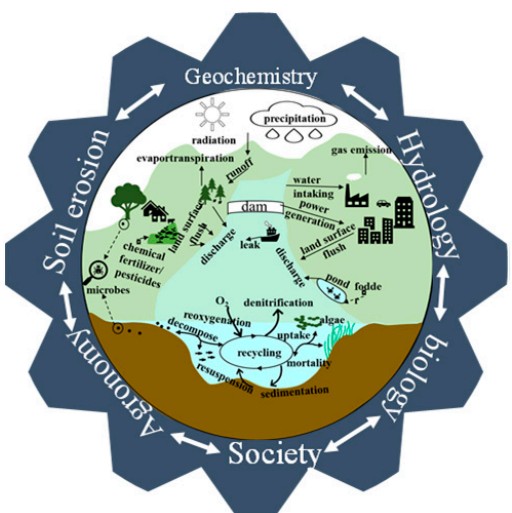

**Figure 1.** Major NPS processes in watershed and disciplines involved (the font size of different disciplines represents their relative development in NPS model research; the solid straight arrows for NPS emission process, hydrological cycle, and water withdrawal by human society, the curved arrows for the biochemical process, the dashed arrows for explanation, the double arrows represent the interrelationship between different disciplines).

The basic simulation processes generally include hydrology, soil erosion, and material transport. Hydrological simulations include infiltration, evapotranspiration, surface runoff, and groundwater processes. Commonly used functions for infiltration include Green-Ampt, Philip, Horton, and Holtan algorithms [16]. The Soil Conservation Service (SCS) is a popular method for runoff simulation and is now generally used in NPS models [17]. Besides, a typical tool for soil erosion is the Universal Soil Loss Equation (USLE) model, which has been modified and extensively used in NPS models [18]. At last, the transportation

of pollutants is simulated based on the runoff processes, the soil erosion process, the morphological transformation process, and the chemical reactions between substances, etc. It is worth noting that it is not always possible to have every detailed process due to simplified framework, special scenarios, and poor data availability.

*2.2. NPS Model Development*

To explore the trends and development of NPS models, we performed a topic search that aimed to capture the maximum possible amount of relevant literature using the Web of Science Core Collection database. We used the terms [TS = (("nonpoint source") OR ("non-point source") OR ("non point source")) AND TS = (model)] as the search queries. The search results were deduplicated, sorted, and irrelevant entries were removed, and finally 3111 relevant documents were obtained as analysis data. CiteSpace, a popular bibliometric analysis software, was first developed by Chen and Song [19]. We used CiteSpace for keyword analysis and then to summarize the development trends of NPS models.

Figure 2 shows the yearly distributions of papers published. It also shows that studies of NPS models began between 1975 and 1990. It is an initial stage for exploration of some processes, such as mineralization of organic nitrogen, ammonia volatilization, carbon transformations. Some conceptual models and NPS models with simple processes are produced based on hydrological models, such as the Chemicals, Runoff and Erosion from Agricultural Management Systems (CREAMS) model developed by the USDA Agricultural Research Service (USDA-ARS), the Groundwater Loading Effects on Agricultural Management Systems (GLEAMS) [20], and the Agricultural Non-Point Source (AGNPS) [21]. Since 1991, publications show a steady upward trend. At this stage, a growing emphasis is being placed on watershed management, pollution control, and water quality improvement, not only in predicting NPS pollution load. The introduction of geographic information system (GIS) facilitated the development of NPS models, and more complex and integrated models were developed, such as the Hydrological Simulation Program-Fortran (HSPF) model [22], Annualized Agricultural Non-Point Source (AnnAGNPS) [23], and a mechanical model for large- and medium-scale watershed management in daily steps, the Soil and Water Assessment Tool (SWAT) [24] developed by the USDA-ARS. The NPS models mentioned above are more related to the terrestrial NPS pollutant transport process, and the conveyance process in rivers is often insufficiently considered. Therefore, some special river and lake models, such as the Environmental Fluid Dynamics Code (EFDC) [25] and One-Dimensional Transport with Inflow and Storage (OTIS) [26], are also widely used for the simulation of the advection and dispersion process of NPS pollutants in rivers and lakes. Models mentioned above are listed in Table 1.

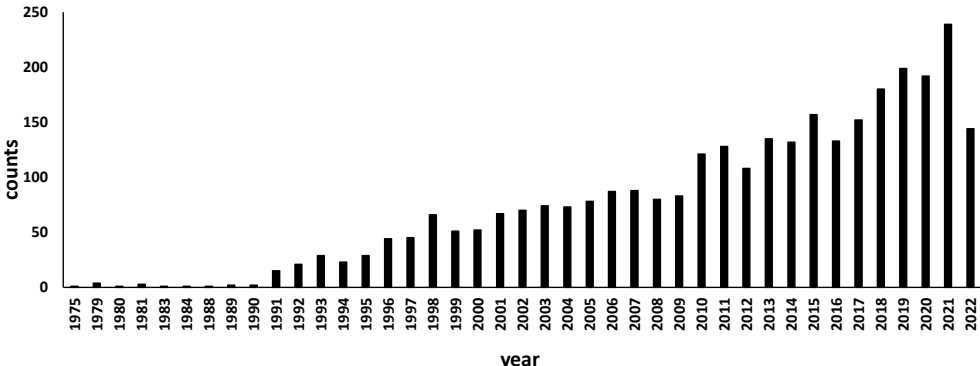

**Figure 2.** Counts of publications of NPS model research by year, Source: authors based on Web of Science database, Term: [TS = (("nonpoint source") OR ("non-point source") OR ("non point source")) AND TS = (model)] (note that 2022 is not yet over, so the collection of 2022 is not complete).

**Table 1.** Comparison of some commonly used NPS pollution models.

| Model | Materials | Spatial Scales | Application |
|---|---|---|---|
| SWAT [24] | sediment, nutrients, pesticide, heavy metal, etc. | watershed | A semi-distributed model that can effectively simulate the spatial and temporal characteristics of runoff and pollutants, and analyze the influence of topographic features, substrate features, land management features, climatic features, hydrological features and other factors on runoff, soil erosion, pollutant leaching and groundwater environment. |
| HSPF [22] | sediment, nutrients, pesticide, salts, pathogens, etc. | watershed | A lumped model that can simulate the runoff and water quality for each subbasin individually, and the migration and transformation process of river pollutants can also be simulated according to the upstream and downstream relationships of the basin. |
| AGNPS [21] | COD, nutrients, etc. | catchment | An event-oriented distributed model that mainly used for agricultural NPS pollution estimation and prediction. It requires less data than other distributed models. |
| AnnAGNPS [8] | COD, nutrients, etc. | watershed | A continuous distributed model that developed from the AGNPS model, inherits the advantages of the AGNPS model except for the simulation of single rainfall event. |
| GWLF [27] | sediment, nutrients | catchment/watershed | A semi-distributed and semiempirical watershed load model; it can be applied to an ungauged watershed, used in estimation of nutrient loads from multiple sources (including point source, rural runoff, urban runoff, groundwater, saprophytic drainage system, etc.). |
| DNDC [28,29] | carbon (C) and N cycles | field | A plot model for simulating C and N cycles in terrestrial ecosystems; it has a good performance in paddy ecosystem; different farm management scenarios can be simulated and analyzed. |
| CREAMS [20] | nutrients, pesticide, etc. | field | A field model to simulate NPS pollution and to simulate the effects of different management practices on pesticide loads in groundwater. |
| GLEAMS [20] | nutrients, pesticide, etc. | field | A field model to evaluate the effect of agricultural management practices on soil erosion, nutrients and pesticide leaching, and runoff with a layering system. |
| APEX [30,31] | sediment, nutrients, pesticide, etc. | catchment/watershed | A spatially distributed model that can divide the field or small watershed area into relatively uniform spatial units according to soil type, landscape location, surface hydrological elements, and management practices; the units are interconnected by river channels. Additionally, it can simulate the contaminants at each subdivision outlet or at the entire watershed outlet. |
| SWMM [27,32] | nutrients, pesticide, TSS, oil/grease, etc. | catchment/watershed | An urban NPS model mainly used for water quality simulation, scenario prediction and pollutant management assessment for single rainfall events or long-term continuous rainfall-runoff processes from primarily urban areas. |
| SPARROW [33] | nutrients, pesticide, heavy metal, etc. | watershed | A regression statistical model of watershed spatial attributes combining empirical statistics and mechanisms; the statistical estimation of land and water parameters separately to quantitatively describe the migration rate of pollutants from source to river and the transport between upstream and downstream of the river network, which is advantageous in terms of data requirement and is also feasible for simulation in areas with uneven distribution of monitoring points. |
| PLOAD [34,35] | nutrients, pesticide, heavy metal, etc. | watershed | A GIS-based model for calculating NPS pollutant loads from different sub-watersheds based on annual or seasonal precipitation and land use. |
| Export coefficient model [36,37] | nutrients, pesticide, heavy metal, etc. | watershed | A typical statistical NPS model that mainly estimates pollution loads based on export coefficients, without considering the influence of subsurface conditions, provides good applicability to areas with scarce data. |
| EFDC [25] | DO, COD, algae, nutrients, active metal, etc. | river, lake, reservoir, estuaries, ocean and wetland | EFDC is a 3D environmental fluid dynamics model, which is widely used in the dispersion process of pollutants in lakes, reservoirs, rivers, etc. This model is often used in the assessment of environmental response to NPS pollution and in the simulation of algal outbreaks. |
| OTIS [26] | water-borne solutes, e.g., chloride, phosphate, nitrate, dissolved metals, etc. | river, stream | A model that focuses primarily on the fate and transport processes of solutes in a river or stream. The model is good at pollutant longitudinal transport simulation but lacks consideration of vertical deposition. Two conceptual areas are defined in the model: the main channel for advection and dispersion processes, and the storage area (the porous area of the dead pool and the riverbed) for transient storage process. |

*2.3. Model Classification*

Model classification helps researchers to quickly understand the differences between models and to help users to choose the most suitable model for a particular problem.

2.3.1. Statistical Model and Mechanism Model

According to methods used to quantify hydrologic processes, NPS models can be classified as statistical models and mechanism models. The statistical model, also named as empirical-based or "black box" model, is based on statistics for describing the functional relationship between different variables, which usually requires considerable data for variables. Since the underlying surface characteristics of the study area are often neglected, it is relatively simple and straightforward and has good applicability in areas with limited data varieties. For example, the export coefficient model, the pollution load (PLOAD), and the Spatially Referenced Regressions On Watersheds (SPARROW) are commonly statistical models. Mechanistic model, also known as the physical-based model, is based on hydrological, chemical, and biological principles and aims to describe the migration and transportation processes of pollutants, the so-called "white box" model. For example, the HSPF, SWAT, AGNPS, Generalized Watershed Loading Functions (GWLF), Agricultural Policy/Environmental eXtender (APEX), and DeNitrification–DeComposition (DNDC) are mechanistic ones. Hydrological processes are the basis for the transportation of NPS pollutants [38], while the migration of pollutants between the water–soil–atmosphere are the main mechanisms [39]. Mechanistic models basically include hydrology, soil erosion, and pollutant transport modules. However, given the complexity of the model structure, the availability of data, and the mechanisms that are not completely clear, "grey box" models are more popular, using a statistical method to determine parameters to quantitatively simulate pollutant behavior under the mechanistic model framework. Empirical approaches such as the USLE, which is acknowledged as a practical tool for predicting soil erosion, have been widely used as soil erosion modules applied in SWAT, AGNPS, and other NPS models [18,21,40].

2.3.2. Lumped, Semi-Distributed and Distributed Model

NPS pollutants behave differently under various combinations of spatial factors such as vegetation, topography, soil, and land use. NPS models usually employ lumped approach, semi-distributed approach or distributed approach for spatial discretization [9]. These approaches express different degrees of spatial heterogeneity and have unequal requirements for input parameters. The lumped (no discretization) model represents a watershed with a fixed set of properties, such as dominant soil, vegetation and land use. The semi-distributed model assigns values to different simulation units based on properties of land use, soil type and topography features. Hydrologic Response Units (HRUs), for instance, used as the fundamental spatial unit, each HRU can be set with unique land use, soil, topography, vegetation and other characteristic parameters [24]. Additionally, it is tendency for HRUs to not interact with each other in a semi-distributed model like SWAT. The distributed model divides a watershed into hydraulically connected elements. Each element has individual parameter set. Such as irregular "cells" of uniform land management and soil used by AGNPS and AnnAGNPS [8,21].

2.3.3. Field-Scale, Catchment-Scale and Watershed-Scale Model

According to the principal processes of NPS pollution at different spatial scales, NPS models can be classified into field-scale, catchment-scale and watershed-scale. Field-scale models focus on water and contaminants transformation processes, especially the vertical material exchange between the soil and atmosphere in agricultural fields, and they rarely account for the connections between upstream and downstream [41]. DNDC is a typical field-scale model. Catchment-scale models allow the study of the interactions between slopes and channels, but the interaction and transformation between various forms of pollutants are not considered enough [42], such as AGNPS. A watershed is

a fundamental unit to simulate various hydrologic, hydraulic, soil erosion, sediment transport, and nutrient dispersion processes that accounts for surface water, groundwater, and their interaction as a whole system [8], the most commonly used watershed models are SWAT, HSPF, AnnAGNPS.

It is important to note that some watershed models are applicable to a very wide range of scales [43]. For example, SWAT model was applied to University of Kentucky Animal Research Center, Kentucky (5.5 km$^2$) [44] as well as Upper Mississippi River basin at Cairo, Illinois (491,700 km$^2$) [45]. Therefore, the scale effect discussed in this review mainly refers to the principal processes of NPS pollution considered in model. We also suggest that researchers choose an appropriate model based on the characteristics of the study area and the issues to be addressed.

### 2.4. Classical NPS Models

#### 2.4.1. The Export Coefficient Model

The export coefficient model is a typical statistical model that is commonly applied to agricultural watersheds. The export coefficient characterizes NPS pollution by establishing a response relationship between pollutant loads generated by different land uses and the intensity of pollution [46]. Since the influence of climate and terrain on NPS pollutant is not considered, its application in large areas is limited. Additionally, it must be carefully "tuned" for local areas, which has big data collection demands. Generally, some modules, such as meteorological factors, terrain factors, and vegetation interception, are often coupled to enhance the availability of the model [36]. For example, the PLOAD model is an improved version of the GIS-based export coefficient that evaluates the annual NPS loads of sub-watersheds [47].

#### 2.4.2. Soil and Water Assessment Tool (SWAT) Model

SWAT is a mechanistic model and is a commonly used semi-distributed watershed model [48–50]. HRUs, originally used in SWAT models, overlap information on land use, soil types, and topographic features like slope to calculate runoff processes, soil erosion processes, sediment, nutrient and other pollutant transport processes in each sub-watershed. The hydrological processes including surface runoff, peak flows, groundwater, evapotranspiration, etc. are simulated based on water balance equations. The soil erosion caused by rainfall runoff can be determined by the Modified Universal Soil Loss Equation (MUSLE). The SWAT model can simulate the transport process of many substances including sediment, nutrients with various forms, heavy metals, etc., which is mainly coupled to hydrological processes [43]. SWAT model incorporates numerous empirical equations for critical parameters or pollutant behaviors. Therefore, it requires not only extensive parameter data, but also a significant amount of measured data of these parameters for development and calibration [51].

#### 2.4.3. Hydrological Simulation Programs Fortran (HSPF) Model

HSPF is also a mechanistic watershed model. It is developed from the Stanford Watershed Model (SWM), adopted and integrated Hydrocomp Simulation Programming (HSP), NonPoint Source Model (NPS), Agricultural Runoff Management Model (ARM) and Sediment & Radionuclides Transport (SERATRA) and traditional hydrological water quality models [22]. HSPF model generalizes a watershed into three modules: PERLND, IMPLND, RCHRES, which can be subdivided into different compartments. The PERLND and IMPLND modules simulate runoff and water quality constituents from pervious and impervious land areas respectively. The RCHRES module is responsible for simulating the hydrology and water quality of the river. ATEM, PWATER, IWATER, SNOW, HYDR etc. compartments in the PERLND and IMPLND modules can simulate the hydrological processes in the watershed; water quality (including sediment) simulations are done with SEDMNT, PQUAL and Agri-Chemical modules, and users can choose different methods depending upon the available data. The simulation algorithms within HSPF, which is

similar to SWAT model, are a mixture of physically-based and empirical approaches. Therefore, it also requires considerable amounts of measured data for calibration and validation. The flexible modular design and robust simulation capabilities make the HSPF model one of the most important tools for water resources management both in urban and agricultural watershed [52–54].

There are numerous NPS models at present, and different models have their own scope of application. Researchers should be fully introduced to these models when making their selection. In this regard, the above analysis, as well as Table 1, provides some suggestions to researchers.

*2.5. Trends of NPS Model Research*

Table 2 shows the top 25 keywords in the literature records of the last 20 years (Source: Web of Science database, Term: [TS = (("nonpoint source") OR ("non-point source") OR ("non point source")) AND TS = (model)]). In addition to "nonpoint source pollution", which is the most relevant search term, "water quality", "land use", "runoff", "nitrogen pollution", "soil", "swat model", and "phosphorus" also appear frequently. It indicates that SWAT is the popular model among the NPS model studies in the past 20 years, meanwhile, nutrients, runoff, and sediment are the main simulated substances. Additionally, the impact of NPS pollution on water quality and watershed management are still the hot topic of research in recent years. The "soil", "land use" and "scale" are some of the major influencing factors that have drawn more attention. "Soil" is the important input parameter for NPS models in terms of soil moisture content, soil type, soil particle size and other attributions. Studies related to "land use" focus on the contribution of different land uses to NPS pollution load and the relationship between land use and water quality. While studies related to "land use change" emphasize predicting the impact of land use change on NPS loads in watersheds. The "scale" includes "catchment", "basin", "watershed" etc. Each "scale" has relatively unique NPS pollution processes, so there are different combinations of "management" and "modelling", such as "watershed management", "catchment modeling", etc. In addition, "uncertainty analysis" and "sensitivity analysis" are also important keywords in modeling studies, which helps to select the appropriate parameter sets for complex models and increases the interpretation of model simulation results.

Unlike the frequency ranking, AGNPS model (2003–2008) has a greater impact in the early years as shows in Table 3. The "geographic information system" has the great impact in early years, which contributed greatly to the development of NPS models. In addition, "watershed management", "flow", "prediction", "spatial variability" and "sediment transport" all have long burst periods, indicating that the spatial heterogeneity of flow and sediment transport, the NPS pollution prediction and the watershed management measures are critical issues at this period. The "hydrological modeling", "watershed modeling", "watershed model" also have a long burst period. Most of NPS models are developed on the basis of hydrological models; therefore, hydrological modeling is an important content of early NPS modeling. The "watershed modeling" and "watershed model" have similar burst periods, but "watershed model" is more concerned with model structure, improving model accuracy, developing new model frameworks, etc., "watershed modeling" emphasizes model application, especially the simulation for nitrogen, total maximum daily load, ecosystem, etc. The "nonpoint source pollution model" has the longest burst period (2003–2014), which corresponds to the distributions of publications, indicating that this is the period of rapid development of the NPS model. From 2005 to 2011, the "total maximum daily load" (2007–2014) and "validation" (2007–2016) have longer burst periods, and "gulf of mexico" have a seven-year burst period from 2011 to 2016, indicating that the Gulf of Mexico was the hot spot. After that, "china" has a short burst period in 2017–2018, indicating that NPS model studies are popular in China since then. In recent years, the study of NPS models has focused on the evaluation and improvement of model performance as well as the land landscape and climate change. Finally, it is

worth noting that the study of NPS models has gradually shifted from "water quality" to "ecosystem service" and the trend is continuing.

**Table 2.** Top 25 keywords with frequency in last 20 years.

| Counts | Year | Keywords |
|---|---|---|
| 1186 | 2003 | nonpoint source pollution |
| 519 | 2003 | water quality |
| 339 | 2003 | land use |
| 319 | 2003 | runoff |
| 316 | 2003 | nitrogen pollution |
| 306 | 2003 | swat model |
| 305 | 2003 | soil |
| 299 | 2003 | phosphorus |
| 267 | 2003 | best management practice (bmp) |
| 216 | 2003 | sediment |
| 199 | 2003 | river basin |
| 199 | 2003 | uncertainty analysis |
| 183 | 2003 | climate change |
| 177 | 2003 | catchment |
| 176 | 2003 | simulation |
| 163 | 2004 | river |
| 156 | 2003 | land use change |
| 127 | 2003 | soil erosion |
| 119 | 2007 | nutrient |
| 116 | 2005 | basin |
| 116 | 2005 | calibration |
| 112 | 2003 | flow |
| 111 | 2003 | scale |
| 108 | 2003 | sensitivity analysis |
| 103 | 2003 | watershed management |

**Table 3.** Top 25 keywords with the strongest citation bursts.

| Keywords | Strength | Begin | End |
|---|---|---|---|
| geographic information system | 14.32 | 2003 | 2008 |
| watershed management | 13.49 | 2003 | 2008 |
| agnps model | 10.1 | 2003 | 2008 |
| flow | 9.01 | 2003 | 2010 |
| water quality | 7.32 | 2003 | 2004 |
| prediction | 5.61 | 2003 | 2012 |
| spatial variability | 5.24 | 2003 | 2010 |
| sediment transport | 5.12 | 2003 | 2012 |
| hydrological modeling | 5.04 | 2003 | 2008 |
| watershed modeling | 4.95 | 2003 | 2012 |
| watershed model | 4.76 | 2003 | 2010 |
| nonpoint source pollution model | 4.42 | 2003 | 2014 |
| watershed | 4.98 | 2005 | 2008 |
| runoff | 4.58 | 2005 | 2006 |
| total maximum daily load | 6.44 | 2007 | 2014 |
| validation | 5.06 | 2007 | 2016 |
| constructed wetland | 4.52 | 2009 | 2014 |
| gulf of mexico | 6.76 | 2011 | 2016 |
| china | 7.59 | 2017 | 2018 |
| landscape pattern | 5.53 | 2017 | 2022 |
| emission | 4.91 | 2017 | 2020 |
| conservation practice | 4.57 | 2017 | 2020 |
| climate change | 9.11 | 2019 | 2022 |
| ecosystem service | 5.23 | 2019 | 2022 |
| performance | 5.02 | 2019 | 2022 |

## 3. Challenges in NPS Model Research

The stochasticity, indirection, and uncertainty of NPS pollution, as well as the scale-variability, make it difficult to simulate. This section summarized challenges in model construction and application (Figure 3).

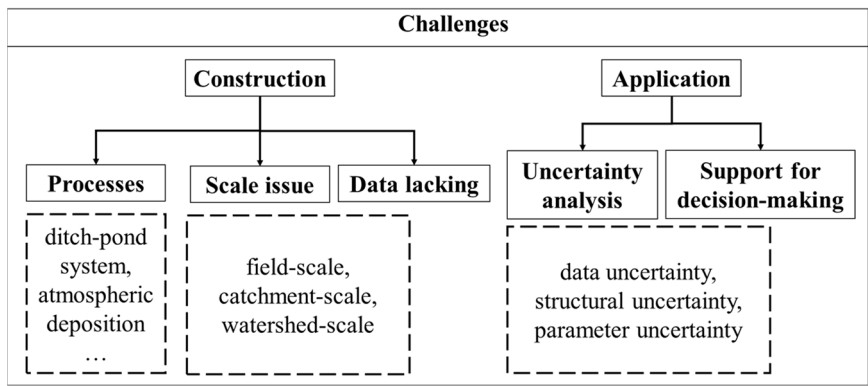

**Figure 3.** Challenges in NPS model research.

### 3.1. Challenges in Model Construction

#### 3.1.1. Specific Process Incorporation

There are many concepts and algorithms of NPS pollution. However, there are barriers that lead to limitations in understanding specific processes.

Firstly, ditches and ponds are important landscapes in rural and agricultural areas. They play a critical role in geochemical cycling [55], and present valuable ecosystem and society functions [56]. However, due to the small size and scattered locations, they are hard to detect on maps or traditional satellite images. Therefore, they are often simplified or ignored in the model, especially in large scale research. Holgerson and Raymond [57] explored the contribution of small ponds (<0.001 $km^2$) to global natural carbon cycling, the results showed that small ponds comprise only 8.6% of the global surface area of lakes and ponds, but account for 15.1% of $CO_2$ emissions and 40.6% of diffusive methane emissions. Authors believe that more research is required on the global distribution of small ponds, and new technologies are needed for small ponds mapping. Currently, few studies have assessed the impact of small ponds on the nutrient cycling at the global scale, but there are many local studies, especially in downstream Yangtze River Basin in China, location of the ditch- and pond-dominated areas. Sun et al. [58] explored the contribution of ditch-pond systems to pollutant removal by developing a model framework. They rasterized natural-artificial drainage channels and realized the accurate spatial representation of ditches and ponds. This framework effectively fills the gaps in the current models regarding the consideration of complex ditch and pond systems. Given that different ditch profiles (soil, concrete, mixed, etc.) and ditch vegetation patterns have a significant effect on the pollutant removal efficiency [59], there is still potential for research on the slow release of pollutants intercepted by ditches and the variation in decontamination efficiency over time.

Second, the detailed source information and transportation of pollutants should be considered. In terms of geochemical processes, there are significant differences in the range and interfacial processes of different pollutants. For example, nitrogen (N) is not only transported between landscape–water interfaces but also readily interacts with the atmosphere in the form of ammonia and nitrous oxide [5]. Traditional research has focused mainly on NPS pollution of the water and soil mass transport interface, with less attention given to atmospheric deposition. The present models of NPS pollution comprise few atmospheric deposition modules, but according to recent studies, the emissions of reactive nitrogen intensify air pollution [60], and N deposition increases watershed N export as a consequence [61]. According to Tian et al. [62], global atmospheric nitrogen deposition is showing a steady increase from 1980s–2016. Chen et al. [6] combined atmospheric deposition data with a distributed NPS model to quantify the effect of atmospheric deposition on

water quality. The results showed the decrease in N deposition ($0.45 \times 10^4$–$1.09 \times 10^4$ T $yr^{-1}$) leads to a decrease in tributary N: P stoichiometry (approaching to 22.04), resulting in a shift from P limitation to N limitation in the reservoir, which is critical for water quality management in the reservoir. Thus, the incorporation of excessive atmospheric deposition and other processes, such as N leaching or gaseous loss, on the biogeochemical cycle should be emphasized; otherwise, the simulation and prediction of NPS pollution will be distorted [63].

Finally, specific biogeochemical processes, such as microorganisms that have a considerable impact on material transport, are less considered at present. For instance, soil N-cycling has a close linkage with soil microbes. Microorganisms play an essential role in the N mineralization, nitrification, and denitrification processes [64]. It has been reported that increased surface temperature can alter soil microbial metabolic processes and rates, which promote N transformation [65]. Given that the consideration of climate change will alter environmental conditions, such as wind speed, precipitation, temperature, etc., as well as microbial metabolic processes, some previously neglected or streamlined processes need to be reassessed for their contribution to NPS pollution for the accurate and precise simulation [66].

### 3.1.2. Model Expansion to Different Scales

Hydrological processes as well as the migration and transportation of pollutants show significant spatial heterogeneity because of the intertwined effects caused by the topography, vegetation, and soil characteristics [58,67].

The field scale mostly considers processes such as soil–vegetation, and the key principle is the knowledge of agronomy and soil science. The DNDC is a typical field-scale model [29]. It was initially used to study greenhouse gas emissions from ecosystems, and denitrification and decomposition are basically the effects of C and N emissions from the soil to the atmosphere. Based on denitrification and decomposition, the model was applied to C and N cycling in different ecosystems. In agricultural ecosystems, the prediction of crop growth, soil C and N dynamics, greenhouse gas emissions and N loss provide a great help to the study of NPS pollution [68].

A catchment scale model would consider the converging processes, for example, precipitation-runoff with the influences of ditches, ponds, and other factors. Catchment models are generally based on hydrology, hydraulics, soil science and many other disciplines. The AGNPS model is a typical catchment model, which is an event-oriented distributed model based on physical processes that can simulate the hydrological cycle, soil erosion, and nutrient loss in a catchment.

Watershed-scale models, such as SWAT and HSPF [69], give more attention to complex river networks, reservoirs, and water extraction for human development based on hydrology, hydraulics, soil erosion and other disciplines.

For field-scale and catchment-scale models, the extension from the field scale to the watershed scale usually divides the simulation area into multiple cells, in which the various conditions are homogeneous. Then, all cells are simulated one by one, and the cells are finally superimposed together. The advantage is that it can divide different cells according to the spatial heterogeneity of the region, and the simulation is more flexible and accurate for each cell [29]. The disadvantage is that it may overlook some transfer processes between cells, leading to biased results overall. Catchment models introduce additional transport channels (e.g., rivers) between cells compared to field scale ones, but catchment models seldom consider groundwater flow; therefore, they remain to be proven for applications in large watersheds. Watershed models often present the strengths of multiple simulation steps, including hourly, daily, monthly, and annual scales, and the spatial scales of simulation include small watersheds, medium- and large-scale watersheds, and even entire regional scales. These features facilitate a detailed analysis of the mechanisms, processes, and environmental impacts of NPS pollution in the watershed [70]. However, considering the complexity and feasibility of the model, some

processes and parameters are generally neglected or weakened. The description of some processes is not as detailed as field and catchment-scale models [49]. In addition, the model structure is complex, and more parameters are needed, providing a great challenge for application, especially in areas with scarce data.

### 3.2. Challenges in Model Application

### 3.2.1. Data Are Still a Problem

Modeling is an efficient instrument in understanding observations and in developing and testing theories but cannot be an alternative to measurement and monitoring. Currently, data scarcity remains a major issue in model applications. Data required for NPS modeling generally include spatial data (DEM, land use, soil type and distribution, etc.), meteorological data (precipitation, temperature, wind speed, solar radiation, etc.), vegetation data, pollutant source data, etc. Given different model structures and research objects, the required data can vary in quantity, size, and resolution [58,71]. Pollution loads can generally be assessed based only on the source data, land use data, and export coefficients by statistical models. In contrast, complex models such as SWAT, HSPF, and AGNPS usually require large amounts of data for delineation of surface features [9,36]. In addition, additional data are needed for special processes (e.g., data of snowfall, melt temperature, etc., for snow module).

On the other hand, although the modernization and promotion of the monitoring network have obtained abundant data (e.g., the hydrological stations collect continuous data on water flow, water level, water temperature, etc.), data monitoring and collection are still difficult in some areas because of various reasons, such as the poor base and the treacherous environment. Even in urban areas where the monitoring networks are well established, obtaining complete and high-precision data is difficult. For example, urban drainage networks, pipe network reconstruction, loss of some original documents, or confidentiality, are still incomplete in some areas. In addition, the matches between data are low due to different monitoring locations, frequencies, strategies, etc., and the integration of data from different sources has become another major challenge [72].

Some complex models can be simplified by using a limited number of modules to reduce input data or using public datasets or developed tools established for running models. For example, there are global databases such as soil data, stream network data, and weather data for models to run. When the accuracy of meteorological data in the study area is not sufficient, the SWAT model also provides the weather generator module to generate the daily data needed for the model based on monthly data [24]. However, if local weather shows any gradients with elevation or distance (usually true in mountain or monsoon-nourished areas), the global sets may not be adequate, and measured data will be better. All of these methods have improved the applicability of models, especially in poor data areas. However, the lack of local data for calibration will lead to a great influence on the interpretation and analysis of simulation results. Based on different sampling frequencies, Piniewski et al. [73] conducted a water quality simulation of a boreal catchment in southern Finland using the SWAT model. The results showed that high-frequency sampling could significantly improve model performance. Additionally, many studies have reported that simulation results respond sensitively to the resolution of input data [74]. For data matching, methods such as interpolation and scale conversion are used to obtain the matching data. Data on flux provide substantial information for NPS pollution processes [38]. However, compared with flow data, the cost of collecting and analyzing water quality is higher. Thus, the sampling frequency is once a month or even less, while the daily water flow can be easily obtained. Therefore, in the flux calculation, it is necessary to use some methods to match the stream flow and water quality data of different scales [75], whereas the simulated value cannot be equal to the real value [76].

The validity and accuracy of the data will directly affect the model performance and the understanding of NPS processes [77]. However, while the techniques of flux monitoring and estimation have been increasing, since the lack of uniform criteria of

these methods and the data derived from different methods can be greatly different, data availability has not largely improved [78]. Therefore, there have been studies on model development and application for data shortage areas, and some studies have investigated the impact of limited data on different methods with a view to providing methodological implications for studies in areas with scarce data [79,80]. At the same time, making full use of satellite remote sensing and advanced scientific technologies to improve the coverage of the monitoring network and obtain more high-precision data is also important in areas with scarce data [81]. In addition, for the consideration of economy and data accessibility in practical applications, how to mine effective information from relatively easy-to-obtain data or how to determine the minimum accuracy under the premise of ensuring the reliability of results is an important problem in model application.

### 3.2.2. Prediction Uncertainty Puzzles Modelers
#### Data Uncertainty

Models require various data but the different monitoring methods, as well as data preprocessing techniques, may lead to large differences in the resolution and accuracy of data. Consequently, there may be great discrepancies in the simulation results, which caused great prediction uncertainty. For example, the resolution of DEM data has a large impact on the depiction of watershed boundaries and topographic slope, which in turn affects the estimation of processes such as surface runoff, sediment transport, and material flow in the watershed [82]. Ground monitoring data, for example, observations of flow, water quality, precipitation, etc., is influenced by the density of monitoring stations, equipment, monitoring frequency, etc. [83]. In areas with sparse monitoring sites, the accuracy of input data is restricted by the method of generating the dataset, and the interpretability of the simulation results is therefore greatly affected [84]. In general, high-resolution input data can improve the performance of the model to some extent, but it is also costly in data acquisition and model running. Shen et al. [85] obtained different resolution data of DEM (30 m $\times$ 30 m, 40 m $\times$ 40 m, 90 m $\times$ 90 m, 200 m $\times$ 200 m), land use–land cover (LULC) (40 m $\times$ 40 m, 90 m $\times$ 90 m, 200 m $\times$ 200 m) by re-sampling and data of soil maps (200 m $\times$ 200 m). They grouped data in all possible ways to obtain twelve combinations as GIS input of SWAT. Additionally, then, they investigated the different effect of these combinations on NPS prediction uncertainty. The result showed that the simulation of sediment increased (3494 ton to 4167 ton) from 30 m $\times$ 30 m to 90 m $\times$ 90 m, and then showed an increase (3659 ton) at 200 m $\times$ 200 m. A similar change could be found by TP outputs. Meanwhile, GIS inputs showed limited influence on runoff and nitrogen (N) predictions. They also proved that there is a threshold of GIS resolution within which more precise data would be less beneficial to SWAT performance. Obviously, determining the appropriate data resolution and obtaining sufficient information with a relatively small amount of data is a very necessary problem for model application.

#### Structural Uncertainty

Modeling is a powerful tool for understanding of observations, and in turn, guided by the understanding of transport processes, NPS models are improved; therefore, the structural uncertainty mainly comes from our limited understanding of the watershed systems. As an example, the simulation of the hydrological cycle is mainly based on the natural processes of evaporation–precipitation–runoff–infiltration in the SWAT. However, Wang et al. [86] believed that the natural water cycle was not sufficient for describing the hydrological cycle in a basin with extensive human activities and proposed the nature–society dualistic water cycle theory. Another example includes legacy nutrients in which there is a consensus that there is a hysteretic watershed response for reduction in nutrient input [87]. However, it is difficult to quantify the magnitudes of legacy nutrients in various environmental compartments or clearly report the timescales of legacy nutrient release into surrounding water bodies [88]. Nutrient input can be accumulated in soil, groundwater, sediment, and other landscapes. A study of the Dengsha River watershed

in China pointed out that excessive levels of P stored in agricultural soils could maintain the growth of crops for over one decade even after P fertilization ceased [69]. The mass–balance function is commonly used for the estimation of legacy nutrients. Since the impact of some processes, such as atmospheric N deposition and denitrification, can hardly be quantified, the results are still uncertain [88]. However, neither statistical models, such as SPARROW, nor mechanistic models, such as SWAT and HSPF, can accurately simulate the legacy nutrient process. In recent years, numerous efforts have been made for accurate and dynamic legacy nutrient assessment, but there is not yet a mature method [89,90].

Additionally, the effect of the input data on the results is also affected by the model structure. Previous research showed that the improvement of model performance by GIS data resolution has a threshold, in which a higher resolution could not perform better than a lower resolution [85]. In fact, the impact of data on the simulation results is determined by the parametric functions. The greater the weight of the input data in the model is, the greater its impact. For instance, the SCS method is not strongly dependent on terrain data; therefore, the GIS data resolution has little effect on its hydrological simulation results. In contrast, the terrain factor required in the modified universal soil loss equation (MUSLE) comes from DEM data directly; therefore, the TP and sediment results are sensitive to the GIS data resolution. Regarding differences in model structure and algorithms, even the same data may pose various influences in distributed, semi-distributed, and lumped models [91–93].

Parameter Uncertainty

Finally, assumptions for NPS pollution processes and generalization of complex processes with limited parameters are important causes of model uncertainty [94]. Although in mechanistic modeling, to account for real environmental conditions as much as possible, there are often many parameters (e.g., >100 in the SWAT model). Due to our limited knowledge, assumptions about a process can lead to parameter selection and setting errors. In addition, facing specific research purposes, not all parameters have a great impact on the simulation results. Therefore, to improve efficiency, parameter sensitivity analysis is required to select specific parameters for simulation and prediction [95]. At present, parameter analysis can be divided into local and global sensitivity analysis. In comparison, global sensitivity analysis can consider the interactions between parameters, which is better than local sensitivity analysis [96]. Neither the local nor the global analysis considers the spatial scale effect of parameters. The obtained parameter sensitivity is constant if sensitivity analysis is based on the whole time series. However, the parameter sensitivity should also be dynamic due to the scale effects of the parameters. There are several methods for the time-varying effect of parameter sensitivity, such as sliding windows and the time-varying and multi time scale (TVMT) method [97]. However, the spatial scaling effect of parameter sensitivity has rarely been reported [98].

Model calibration and validation are important steps in model application, reflecting the analysis of model uncertainty, which supports the interpretation of simulated results [50,99]. However, the traditional model evaluation only considers the goodness-of-fit between the measured and predicted data [100], which is not sufficient for model uncertainty. The confidence interval (CI) and a probability density function (PDF) have been proposed to express the uncertainty of model prediction [101]. Chen et al. [102] proposed an interval-deviation approach (IDA) method for model evaluation in an uncertainty framework. However, this approach is suitable for situations where there are less data, and these intervals may not always be feasible when more data can be collected or when a continuous and random data distribution can be assumed. The cumulative distribution function approach (CDFA) and the Monte Carlo approach (MCA) are commonly used methods for model evaluation in uncertainty frameworks [103]. Nevertheless, due to limited knowledge and natural randomness, a fixed PDF or error range cannot be found. Thus, researchers should be more cautious about the evaluation of the error range of measured data that can be used in watershed simulation. More data should be collected to obtain a

reasonable margin of measurement error and an appropriate PDF of the predicted data. In addition, due to scarce data, measured data and a model and its calibration are for a small part of a catchment, often a single drainage, but the use of the model is scaled up to cover the entire basin on the basis of one or two parameters without any assessment of whether that scaling is justified by the distributions of measured and modeled data.

### 3.2.3. Insufficient Accuracy for Decision-Making

Watersheds are important spatial units of water environmental governance and a key research scale of pollutant transport. BMP or low-impact development (LID) practices are important means for watershed management, and many NPS models, such as SWAT, HSPF, AnnAGNPS, APEX, and Storm Water Management Model (SWMM), have integrated the BMP/LID module for watershed management [4,104]. Watershed models can evaluate the impacts of multiple BMP on watershed hydrology and water quality. For structural BMP (e.g., filter strips, riparian buffers, and detention ponds), some specific models have been developed, such as the Vegetative Filter Strip Model (VFSMOD) and Riparian Ecosystem Management Model (REMM) [105,106]. These specific models require higher resolution data compared to watershed models. Therefore, they are more suitable for farmlands or small watersheds with fully functional databases.

There are some problems in BMP module with models. First, there is a mismatch between the simulation unit of the watershed model and the BMP unit [4]. For example, users would assess BMP at watershed and sub-watershed scales using the SWAT model, and the hydrological response units (HRUs) may represent the field-level conditions of the BMP, but there is a clear disconnect between the hydrological scale of the HRU and the actual field where these BMP are implemented. Second, before conducting BMP evaluation, several steps, such as model calibration, validation, and simulation, must be completed, which is tedious and not easy for managers. However, some tools, such as the Pasture Phosphorus Management (PPM) calculator and Texas BMP Evaluation Tool (TBET), can hide the complex model simply as an engine, helping users to simplify the model and improve its operability [107,108]. However, the essence of the model remains unchanged, and the accuracy of the simulation results is still affected by many of the factors mentioned above. If managers do not understand the principles, it is difficult to interpret the simulation results when the predefined situation changes. Finally, the largest problem with the evaluation of BMPs in both watershed and specific models is that they rarely take into account the decreasing efficiency of BMP over time. Bracmort et al. [109] assessed the efficiency of grassy waterways, grade stabilization structures, field borders, and parallel terraces in reducing sediment and TP using the SWAT model. The temporal variability has been considered. The results showed that the BMP in good condition (fully functional) reduced sediment up to 32% and phosphorus up to 24%, while degraded practices (partially functional) reduced sediment by only 7% to 10% and phosphorus by only 7% to 17%. Thus, current models may fail for future watershed decision-making. Therefore, a decay function would be helpful for evaluating varying efficiency of BMPs; in addition, factors such as maintenance activities and pollutant accumulation can also greatly affect the life of BMPs, which should be considered carefully.

## 4. Future Prospects for NPS Model Development

Modeling is an approximation of reality and will inevitably be distorted; it is important to recreate the real situation as much as possible and improve the precision and accuracy. There are still things to be done in this respect, including but not limited to the following aspects.

### 4.1. The "Source–Flow–Sink" Framework for Model Development

To better integrate mechanistic research from different disciplines, a new framework is proposed for the development of NPS models based on the "source–flow–sink" process (Figure 4). The principles of different disciplines are utilized as three modules, "source–

flow–sink", where the "source" module strengthens the source information of pollutants in farmland, urban, mining, forestry, rural, livestock, and farming. Meanwhile, the actual measurement data serves as a strong support for modeling, the pollution source data such as: source type, amount of pollutant generation, emission etc. are very important. The "flow" module solutions for different regional production and convergence studies strengthen the role of atmospheric deposition and the role of groundwater, coupling more human activity impacts. In this part, the collection of data such as rainfall, river flow etc. is critical for modeling. The "sink" module focuses on wetland, vegetation, ditch, reservoir, river, and lake module development, especially the impact of policy-driven vegetation restoration and reservoir construction. For this module, data like water qualities of rivers, and reservoirs etc. are essential for model calibration and verification.

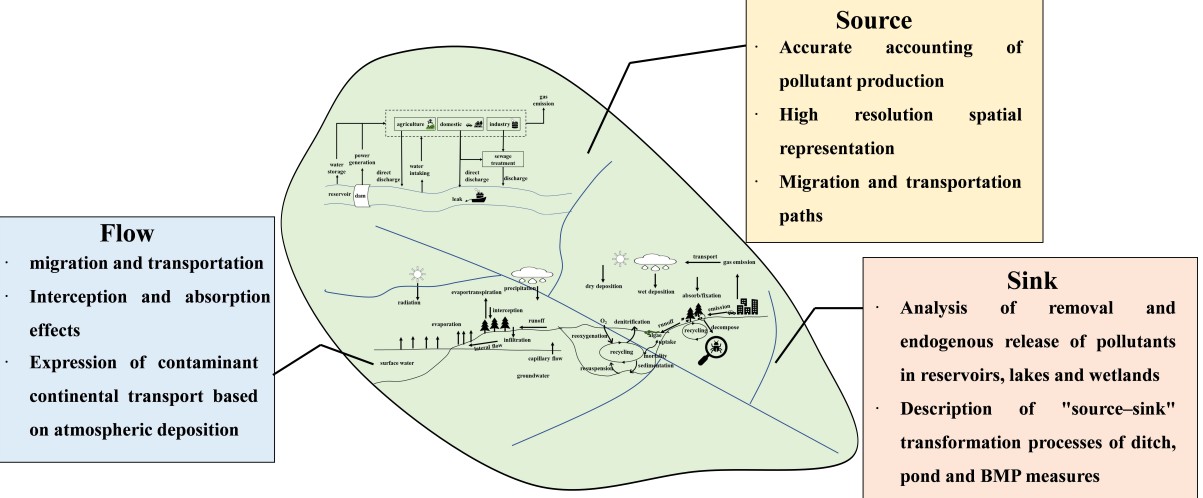

**Figure 4.** The main points in "source–flow–sink" modules.

With global climate change, the "source–flow–sink" relationship will change. However, current studies have only predicted the change in a certain environmental factor, and there is significant interaction between these factors. The enhancement of one factor may weaken or promote the effect of other factors, resulting in more complex and difficult-to-predict pollutant migration and transportation in the basin. There is no doubt that much work needs to be done to study NPS pollution mechanisms. This paper proposes a new "source–flow–sink" framework, based on which of the various paths and processes of NPS pollutant transportation should be identified and made conducive for multi-process development and integration. This may be a promising direction for model development.

*4.2. Standardization for Model Input and Parameter*

Many models have been developed in recent years, but the complex structure of these models is an essential barrier for environmental managers to use. The statistical model is relatively simple, but it is difficult to meet the integrated management of watersheds. In addition to the complex structure of the mechanistic model, the model structure also varies greatly, as do the parameters of the model and the required input data. Therefore, it is necessary to standardize NPS models to address complex watershed circumstances.

First, a general NPS model should include a hydrology module, meteorology module, erosion module, and pollutant transport module for terrestrial and aquatic areas (including surface water and groundwater), and some models should also include a hydrodynamic module (e.g., HSPF). Standardizing the representation of modules allows users to select a suitable combination of modules for different purposes. This will notably minimize scale effects and the obstacle of model coupling and solve the problem in which the current comprehensive model is too complicated to apply, while the simple model has poor interpretation of the results.

Second, a standard database should be established. An ever-improving monitoring network and a variety of portable devices provide us with access to massive data. The standard database of input data that ensures a combination of different modules will work successfully and will then improve efficiency in model coupling, model improvement and application, especially in areas with scarce data. In addition, calibration and validation are complex and massive, and it is beneficial for efficiency to construct a standardized parameter bank. The bank in which all parameters required for modeling can be satisfied will enable researchers to handle different models simply by learning this parameter bank. Additionally, it is useful to strengthen the comparison of model parameters performed by different studies in a region to reduce its uncertainty.

Finally, standard calibration can incorporate manual or automatic processes. Manual calibration relies on the practical experience of the user to adjust the simulation results close to the actual measured values, which is more effective but less efficient. In regard to a particularly large numbers of parameters (e.g., SWAT model), this will cost enormous time and effort. Automatic calibration is comparatively efficient but can easily result in parameter values that do not match the actual situation. Many integrated models generally offer both manual and automatic modes, and users can choose a single mode or combination of the two to improve efficiency. However, there are still some models that are not adapted to the automatic calibration platform (e.g., HSPF model), and for these models, the calibration platforms of other models are not necessarily applicable. Therefore, it is important to develop a common calibration platform with good optimization algorithms. In addition, this is also beneficial for multi-model comparison. By the way, the purpose of calibration is to make the model output closer to the measured data, and we have to note that model results are not equivalent to measured data; those who use models must be absolutely clear to distinguish between measured and model results.

### 4.3. Make Reliable Decision Support

Carrying out multi-model comparison and developing a platform for comparison will be convenient for model users to choose a suitable model and come up with reasonable environmental management strategies. Based on the mechanistic study and multi-model comparison, the key parameters of the model are modified, and the relevant processes (often described by empirical and semiempirical formulas) are improved. For local studies, it will help to localize the model and make the study more focused, the exploration of local mechanisms can promote the improvement of the model, and the performance for large-scale simulation can be better. Models have different characteristics due to scale effects.

Developing and improving more modules, such as atmospheric deposition modules, microbial metabolic process modules and anthropogenic modules, might be an important trend. Since some models have been developed to be relatively mature, albeit containing limited modules, the integration or coupling between models would be a promising trend. For example, the coupling of SWAT and Modular Three-Dimensional Finite-Difference Ground-Water Flow Model (MODFLOW) can more accurately reflect surface and groundwater interactions [49]. Given the feedback between the models, such as social and economic development and the hydrological cycle, two-way coupling would be more reasonable than one-way coupling [39]. For models based on different operating platforms (Win/Linux), different data formats and resolutions increase the difficulties in model coupling. Therefore, it would be a great issue to select an optimum scale transformation method or develop an efficient one [110]. In addition, as models evolve, either seamlessly integrated models that combine multiple functions and processes or the model collection platform where multiple models are closely combined require powerful computer algorithms and data-processing capabilities.

## 5. Conclusions

In recent years, NPS models have constantly improved in regard to varying environmental conditions. However, with global climate change, local weather, hydrology and vegetation will change in the watershed, which will propose challenges for NPS model. In this review, we try to analyze some challenges that exist in the current NPS models. The unclear mechanisms of some modeling processes due to disciplinary barriers, model limitations of spatial heterogeneity, lack of high-quality data, prediction uncertainties and insufficient accuracy for decision-making are discussed in this paper. On the one hand, the poor understanding of some NPS pollution processes leads to limitations in the model structure, which is one of the main reasons for the uncertainty of the model predictions. On the other hand, the spatiotemporal heterogeneity of NPS pollution requires significant amount of measured data for model calibration and validation, but the scarcity of data makes this very difficult. Although models cannot replace actual measurements, NPS models serve as powerful tools for understanding NPS pollution processes, and we still expect them to provide sufficient support for decision making. Therefore, we sorted out the whole process of NPS pollution based on our understanding and proposed a "source–flow–sink" conceptual framework, which, notably, is not a specific model. This framework aims to provide a new perspective for model researchers on NPS pollution process, then to guide model improvement. In addition, the concept that standardization for model input and parameter and making reliable decision support are also be proposed. Anyway, limited by our own research experience and knowledge, we can only summarize the issues that we have encountered, and propose our suggestions in the hope that they will inspire researchers in this field.

**Author Contributions:** Writing—original draft preparation by M.W.; writing—review and editing by L.C., L.W., L.Z., H.X. and Z.S. All authors have read and agreed to the published version of the manuscript.

**Funding:** This research was funded by the National Key R&D Program of China (No. 2021YFD1700600), the National Natural Science Foundation of China (No. 42277044), and the Fund for Innovative Research Group of the National Natural Science Foundation of China (No. 52221003). The APC was funded by the National Key R&D Program of China (No. 2021YFD1700600).

**Institutional Review Board Statement:** Not applicable.

**Informed Consent Statement:** Not applicable.

**Data Availability Statement:** All analyzed data in this study has been included in the manuscript.

**Conflicts of Interest:** The authors declare that they have no known competing financial interests or personal relationships that could have appeared to influence the work reported in this paper.

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
