# Peer review of "Review of Nonpoint Source Pollution Models: Current Status and Future Direction"

_water, doi:10.3390/w14203217_

Round 1
Reviewer 1 Report (Previous Reviewer 3)
Introduction
This revised paper reviews models used in the study of non-point sources (NPS) of water pollution and has added some interesting information about citation and word frequency for the models.. The focus is certainly an important area of research and public need. I continue to believe that the paper could have more impact in these important areas of research and public policy if it contained more quantitative and specific examples, but the authors have chosen to stay with a text-rich approach.
General comments that should be addressed
1. In the introduction and elsewhere you must define, flexibly, the size of the different sorts of areas you are discussing. Terms like “watershed” and “catchment” and “field plot” have different definitions in different countries, for different professions, and probably for different managers. This must be done.
2. Many of the shortcomings of previous versions of the manuscript have been addressed and addressed well. However, there are still translation issues that result in sentence fragments in several places and challenges with agreement between subject and verb. I have pointed many of these out, but I did not find them all.
3. I believe that the paper will now represent a useful contribution to the literature of NPS models and their challenges.
Specific line-by-line comments and suggested corrections are attached as notes and suggested corrections to the draft manuscript

Author Response
Please see the attachment.

Reviewer 2 Report (New Reviewer)
The author has made detailed revisions to the article, the manuscript can be accect with some further revisions:
1. It is mentioned in the abstract that this paper mainly includes three aspects: one is the description of the model framework and some commonly used models; Second is the challenges in model structure and applications; Third, is the future directions of NPS models. However, the model framework and model classification are introduced in the first part of the introduction, and the model development and common models are introduced in the second part. As a result, the structure of the abstract and the article is inconsistent, the introduction part is too long, and the focus is off. Therefore, it is suggested to adjust the contents of sections 1.1 and 1.2 of the introduction to the second part for further integration.
2. In the section on Model Classification in 1.2, the model classification is generally described from two aspects, one is from the aspect of statistical models and mechanism models, and the other is from spatial discretization. Therefore, it is recommended to use the title for grading or add corresponding association words to make the structure clearer.
3. In section 1.2 Model Classification, the full name of the HSPF, SWAT, and AGNPS models should be indicated when the abbreviation of the model appears for the first time
4. Adding corresponding diagrams in the third part (Challenges in NPS model research) is suggested to help understand the challenges of NPS model research.
Author Response
Please see the attachment

This manuscript is a resubmission of an earlier submission. The following is a list of the peer review reports and author responses from that submission.
Round 1
Reviewer 1 Report
Dear Authors,
The amount of work that went into preparing this manuscript is amazing. However, there are just too many statements that are incorrect. I indicated a few in the manuscript.
My recommendation is to scale down this manuscript from worldwise to the area they are familiar with, discuss how the various NPS models have performed, and then come up with realistic recommendations on the suitability of the model for this particular area.
It should not be that much work to change the emphasis of this manuscript,
Detailed comments are given in the attached file

Reviewer 2 Report
Dear Authors,
The paper is scientifically sound, while there is ample scope for improving the language and reducing text. I have made several comments (22 in number), found in the attached pdf file. I hope this will further improve the paper.

Reviewer 3 Report
Introduction
This paper reviews some of the models used in the study of non-point sources (NPS) of water pollution and contains local hints that the authors are most familiar with and interested in nitrogen and phosphorous. The focus is certainly an important area of research and public need. However, as I read the manuscript and read it again, I think it could have more impact in these important areas of research and public policy if it contained more quantitative and specific examples—ideally displayed as figures. As presented, the manuscript contains many lines of text, but only two conceptual figures, one table and little discussion of actual data. More extensive use of graphics and specific examples would help to better inform readers. Many Water readers are interested in but not wholly familiar with models, model function, performance and challenges. The authors do refer readers to a considerable body of literature, but showing plots of results or summary tables, perhaps focused on a couple of specific parameters (nitrogen or TP?) will help the reader understand and keep them interested.
General comments that should be addressed
1. In most cases modelers writing about their work fail to distinguish between measured values and those they model or simulate......and come to believe that model results are "measured". This issue can be overcome, in part, by clear writing, by showing the actual datasets, and by statistically valid comparisons to assess model performance. The authors would be successful if they were able to convince even a few modelers to “show” the data.
2. In the introduction and elsewhere you need to define, flexibly, the size of the different sorts of areas you are discussing. Terms like “watershed” and “catchment” and “field plot” have different definitions in different countries, for different professions, and probably for different managers.
3. I think that the bulk of the manuscript would be greatly improved if the authors gave the reader specific examples. You seem interested in N and NO3-N and you also note studies that focus on P. Perhaps you can use data from your references and own research to discuss, using graphs and tables, actual case studies so your reader can understand the quantitative effects of these different modeling approaches. Are the simulation data 10% from measured? 50%? 100%? You might also discuss, at least qualitatively, the costs vs benefits of additional monitoring at different scales.
4. Common to almost everything you discuss is a continuous record of hydrologic (and meteorologic) data, particularly measured flow for different parts of the study catchment. Second most important seems to be concentration data for the NPS pollutants of interest such as N and P forms. It would really help your readers to give some sort of examples of how models work and can be tested using two specific constituents such as water flow and N concentration. I have to assume that meteorologic data are widely and consistently collected
Specific line-by-line comments and suggested corrections are attached to the draft manuscript

Round 2
Reviewer 1 Report
Please see the attached comments

Reviewer 3 Report
Revisions by the authors have improved the paper considerably, though there are still minor errors. The paper would have much broader appeal if it:
1. Included illustrations or graphs of some sort
2. Chose specific NPS pollutants (sediment? N? P?) and then used them in specific examples in the text. The authors do some of this now but not nearly enough. For most readers graphic illustration of, for instance, N-based pollution, would help them visualize what is discussed in the text.
I worked hard to suggest changes in the abstract that would help the presentation.....and have made minor notes in the attached .pdf, but did not review it in depth.
